# Graded 2D/3D Perovskite Hetero-Structured Films with Suppressed Interfacial Recombination for Efficient and Stable Solar Cells via DABr Treatment

**DOI:** 10.3390/molecules28041592

**Published:** 2023-02-07

**Authors:** Muhammad Mateen, Hongxi Shi, Hao Huang, Ziyu Li, Waseem Ahmad, Muhammad Rafiq, Usman Ali Shah, Sajid Sajid, Yingke Ren, Jongee Park, Dan Chi, Zhangbo Lu, Shihua Huang

**Affiliations:** 1Provincial Key Laboratory of Solid-State Optoelectronic Devices, Zhejiang Normal University, Jinhua 321004, China; 2Division of Science and Technology, Department of Physics, University of Education Campus Dera Ghazi Khan, Multan 32200, Pakistan; 3Institute of Biomedical Materials and Engineering, College of Materials Science and Engineering, Qingdao University, Qingdao 266071, China; 4Department of Physics and Astronomy, University of Florence, Via Giovanni Sansone 1, I-50019 Sesto Fiorentino, Italy; 5Department of Chemical & Petroleum Engineering, United Arab Emirates University, Al Ain P.O. Box 15551, United Arab Emirates; 6Hebei Provincial Key Laboratory of Photoelectric Control on Surface and Interface, College of Science, Hebei University of Science and Technology, Shijiazhuang 050018, China; 7Department of Metallurgical and Materials Engineering, Atilim University, Ankara 06836, Turkey

**Keywords:** DABr-treatment, diethylammonium bromide, 2D capped 3D perovskite, resilient, surface treatment

## Abstract

Several strategies and approaches have been reported for improving the resilience and optoelectronic properties of perovskite films. However, fabricating a desirable and stable perovskite absorber layer is still a great challenge due to the optoelectronic and fabrication limitations of the materials. Here, we introduce diethylammonium bromide (DABr) as a post-treatment material for the pre-deposited methylammonium lead iodide (MAPbI_3_) film to fabricate a high-quality two-dimensional/three-dimensional (2D/3D) stacked hetero-structure perovskite film. The post-treatment method of DABr not only induces the small crystals of MAPbI_3_ perovskite secondary growth into a large crystal, but also forms a 2D capping layer on the surface of the 3D MAPbI3 film. Meanwhile, the grains and crystallization of 3D film with DABr post-treatment are significantly improved, and the surface defect density is remarkably reduced, which in turn effectively suppressed the charge recombination in the interface between the perovskite layer and the charge transport layer. The perovskite solar cell based on the DABr-treatment exhibited a significantly enhanced power conversion efficiency (PCE) of 19.10% with a notable improvement in the open circuit voltage (*V_OC_*) of 1.06 V and good stability, advocating the potential of this perovskite post-treatment approach.

## 1. Introduction

The emergence of organometallic halide perovskites has attracted significant attention from the photovoltaic research community recently as a novel light-absorbing material [1,2]. The remarkable leap of perovskites as emerging photovoltaic materials is attributed to the certain exceptional features and a number of their amazing properties, such as appropriate optical band gap, long-range charge transportation and excellent light absorption features [3,4,5,6,7]. Therefore, solar cells based on the perovskite absorbers have seen a rapid enhancement in power conversion efficiency (PCE) from 3.8% to 25.7%, [6,7,8,9,10,11]. according to the optimized material stoichiometry, solvent engineering, sequential deposition technique, [12,13,14,15,16,17,18,19]. interfacial engineering and additive engineering [20,21,22,23,24,25,26].

The standard MAI-based perovskite thin film fabricated through the fast and facile anti-solvent-induced deposition method is prone to degrade under exposure to moisture, light and high temperature [11,17,19]. Some studies indicate that large grain and good crystallinity of perovskite film can inhibit this degradation [20,21,22,23]. In order to slow down the process of grain growth for good crystallinity, anti-solvents such as chlorobenzene and methylbenzene are introduced, which are beneficial for achieving good morphology of perovskite film [4,11,18]. Although the anti-solvent can produce films with optimum grain size and reduced grain boundaries, this technology cannot suppress the surface defects and moist instability of perovskite film [15,16,17].

In the past few years, two-dimensional (2D) perovskites which are an additional group of perovskite active layers have been extensively used in solar cells, owing to their inimitable structural properties in contrast with the pristine perovskite [10,16,25]. 2D perovskites consist of long-chain organic cations which render these materials excellent resistance to the moist environment [16,17,18]. It is reported that introducing 2D perovskite onto the 3D-based perovskite material has been considered a novel way to improve the long-term stability of solar cells without sacrificing the photovoltaic performance [19,26]. The notion of forming a 2D material capping layer on the surface of 3D films is also reported frequently, resulting in the improvement of the long-term stability and open-circuit voltage (*V_OC_*) for PSCs [24]. Smith et al. reported (PEA)_2_(MA)_2_Pb_3_I_10_ (PEA = C_6_H_5_(CH_2_)_2_NH_3_, MA = CH_3_NH_3_)-based 2D perovskite solar cells (PSCs) with PCE of 4.73% in 2014 [27]. Lately, Tsai et al. reported an enhanced PCE of 12.51% for 2D PSCs using (BA)_2_(MA)_2_Pb_3_I_10_ and (BA)_2_(MA)_3_Pb_3_I_13_ (BA = CH_3_(CH_2_)_3_NH_3_) perovskite films, and the devices showed improved stability under light and humid conditions [15,28]. Yao et al. employed a 2D/3D bi-layered structure to improve both the efficiency and the stability of triple cation PSCs, which incorporated n-propylammonium iodide PAI in a 3D perovskite [29]. Hu et al., for the first time, combined 2D perovskite (PEA)_2_(MA)_4_ Pb_5_I_16_ with 3D perovskite MAPbI_3_ to form a 2D/3D hetero-structure. They showed that the 2D/3D approach could boost the performance and stability of 3D PSCs [30]. However, some long-chain organic cation halide-treated perovskite not only suppressed the charge transfer rate but also reduced the carrier collection efficiency of the charge selective layers at the interfaces, due to the poor electrical conductivity of the long-chain organic cations. As a consequence, inferior performance was observed in 2D PSCs compared with conventional 3D devices [10,14,29]. Therefore, it is needed to explore new suitable long-chain organic cation halides to treat 3D methylammonium lead iodide (MAPbI_3_) film for enlarging the grain size, passivating the defects, and optimizing the device performance [17,30,31]. It is further interesting to form a 2D capping layer over the top of 3D perovskite to entertain as an interfacial layer, which can enhance the device’s resilience without compromising performance. Diethylammonium bromide (DABr, DA = (CH_3_CH_2_)_2_NH_2_) has the same cation as diethyl-ammonium hydrochloride (DACl) and is the isomer with butyl-ammonium bromide BABr [31,32,33,34,35]. Therefore, DABr is a promising choice for treating MAI-based films to enhance the crystallinity, enlarge the grains size, and form a 2D perovskite layer capped on a 3D MAPbI_3_ layer, thus enhancing the V_OC_ and stability of the device [33,34,35,36].

Herein, we introduced a simple but novel approach to fabricate 2D/3D-stacked PSCs by using DABr to treat the surface of MAPbI_3_ perovskite films. The 2D perovskite was synthesized on top of MAPbI_3_ film to form an efficient and stable absorber layer. Our results indicate that the device with a 2D perovskite capping layer results in an obvious boosted fill factor (FF) and *V_OC_*. Compared with the pristine 3D perovskites, 2D-capped 3D perovskite exhibited even morphology with larger grain size and uniform layered structure with fewer pinholes, enabling the absorber film to enhance moisture tolerance. Photoelectric tests of this study demonstrated that the post-treatment of DABr enhanced the charge extraction and suppressed trap-assisted recombination in MAI-based PSCs devices. As a result, the optimized devices achieved a high PCE of 19.10% and long-term stability.

## 2. Results and Discussions

Herein, we introduced a facile but novel halide perovskite material of DABr to treat pre-deposited MAPbI_3_ films for achieving 2D/3D-stacked perovskite architecture. The developed technique involved the DABr post-treatment of one step MAI-based film and followed by brief annealing. DABr not only directly reacts with surplus of PbI_2_ in the MAI based film, but also interacts with MAPbI_3_ via organic-cation exchange to form 2D-capped 3D MAPbI_3_ hybrid film [25,37]. 2D/3D perovskite films exhibited excellent surface morphology with large grain size dimension, good crystallinity and notable uniformity. The 2D capping layer can effectively passivate the surface defects of 3D perovskite layer, leading to suppress the interfacial non-radiative recombination and boost the hole extraction, thus enhancing the *V_OC_* of the device [20].

Figure 1a schematically illustrates the stepwise experimental processes for preparing highly crystalline, large-grain DABr capped MAPbI_3_ thin films via DABr post-treatment with altered DABr concentrations viz. 1.5, 2.5, and 3.5 mg·mL^−1^ in 2-isopropyl alcohol (IPA) solution. During the second stage of the spin-coating process, 100 µL of anhydrous chlorobenzene (CB) was dripped onto the center of the rotating film 10 s prior to the end program. As shown in Figure 1b, the surface morphology of the as-deposited MAPbI_3_ perovskite film was optimized with a high-quality uniform top appearance by DABr treatment. It is inferred that the introduced DABr interrupts the centrifugal progress of MAPbI_3_ and leads to a uniform expansion in all dimensions. The post-anneal for the MAPbI_3_ film with DABr treatment causes the perovskite grains to grow significantly again, resulting in the increased crystallinity of the final perovskite films [25,34].

The evolved changes in surface morphology of the prepared 2D-capped 3D perovskite films before and after the DABr surface treatment were explored through scanning electron microscopy (SEM) images, as shown in Figure 2a–d. The calculated grain size distribution of the respective treated films is summarized in Appendix A. It was previously reported that the morphology of the perovskite film with a large grain size and a compact even surface is a critical factor to attain high photovoltaic performance [18]. The orthodox control film, as expected, showed irregular grain morphology with small average grain size distribution within the proximity of ~200 nm (Figure 2a and Appendix A). Interestingly, when the deposited films were treated with varying concentrations of DABr, changes occurred in both the surface morphology and the grain size. The morphology of perovskite film treated with 1.5 mg·mL^−1^ DABr exhibited reasonable improvement in surface uniformity and compactness with a little surge in grain size, where the average grain size distribution is reported within the proximity of ~450 nm (Figure 2b and Appendix A). The DABr-2.5 film showed large grain evolution and in uniformly packed symmetry, with grain size larger than approximately ~600 nm, as witnessed in Figure 2c and Appendix A. However, the grain quality of the film deteriorated as the concentration of DABr further increased to 3.5 mg·mL^−1^ in solution (DABr-3.5), despite the fact that the grain size distribution remained approximately ~620 nm (Figure 2d and Appendix A). Remarkably, the increased grain dimension of the perovskite film with DABr-2.5 could be caused by the newly formed layer that homogenously covered the control film, which is speculated to be the 2D perovskite capping layer. Therefore, the large grain 2D capping layer with fewer grain boundaries can suppress the defects of perovskite, resulting in inhibiting the non-radiative recombination and enhancing the device’s performance [38,39,40]. The SEM images confirmed that surface treatment with a suitable small amount of DABr can promote a homogenous film crystallization, leading to good quality 2D-3D-stacked films with large grains and homogeneous surface morphology. The witnessed gradually and significantly suave film treated with DABr-2.5 confirms that 2.5 mg·mL^−1^ is the optimal concentration for DABr to improve the surface morphology of perovskite film.

To ponder into the nexus of DA and Br onto the pre-deposited film, 3D-MAPbI_3_ films were treated with Br halide in combination with different DA alternate ammonium cations (BA and PEA). The SEM images of the resultant BABr-2.5 mg·mL^−1^ and PEABr-2.5 mg·mL^−1^ capped films are shown in Appendix A. PEABr- and BABr-treated films parade smaller grain compared with DABr-2.5 mg·mL^−1^-treated film (Figure 2c). Consequently, it was seen that the surface morphology of the films post-treated with different ammonium salts was determined to the type of organic cation. The DABr performed impressively in this case. Appendix A shows the molecular formulas and chemical structures of DABr, BABr and PEABr, depicting the position of the ammonium molecule in the chain network. To further explore that whether only Br that can effectively induce the emergence of 2D layer over MAPbI_3_ film, the DA in combination with other halides are also applied to treat the surface of perovskite film. The SEM images of MAPbI_3_ films treated with different DA halides are recorded, as shown in Appendix A. The grain size of all DA halides-treated films surged notably than that of the control film, which directs that all the halides perform better with DA cation in covering the 3D-MAPbI_3_ films, but DABr stood exceptionally. With the same doping concentration for the DA cation, the grain size of the DABr-treated film is comparatively larger than that of the DAI and DACl-treated films with homogenous surface coverage. All the results indicate that both DA cation and Br anion are involved in the induced merger with MAPbI_3_ grains during post-treatment. It was reported that this large homogeneous crystal grain formation of DABr-treated film could be attributed to the lower surface energy of DABr-treated perovskite [25,37,41]. The DABr-2.5 solution primarily dissolved in the small crystalline owing to the higher surface energy, whereas the second step involves the formation of Br-related large crystalline grain. The above results indicated that the 3D MAPbI_3-_based perovskite layer treated with 2.5 mg·mL^−1^ DABr can induce secondary growth for the control film which incorporates with small size grains, thus turning them into larger grain size 2D-3D stacked films [34].

The crystalline phase and properties of the DABr-treated and control perovskite films were analyzed via an X-rays diffraction (XRD) pattern. Figure 3a shows the XRD patterns of 2D-3D based films with numerous DABr concentrations. As expected, MAPbI_3_-based film shows the prominent diffraction peaks of the pure phase tetragonal MAPbI_3_ with moderate peak intensity [20,39,40]. For the DABr-treated films, the gradually strengthened pattern with the characteristic peak at ≈14.4° shows a little shift towards the higher degree planes. As shown in Figure 3b, all three dominant peaks of the DABr-treated films are slightly shifted to larger diffraction angles, perhaps due to the incorporation of small radius Br ions into the base 3D MAPbI_3_ layer [38,39,40,41]. As shown in Figure 3a, the peaks under 10° are observed in the XRD patterns of DABr-treated perovskite films, which is the evidence for the appearance of 2D perovskite phase [35,39,40]. It is noticeable that the peak intensities of XRD pattern increase as the perovskite layer treats with DABr and the peak intensity rises up to the highest in DABr-2.5-treated perovskite film, which indicates the enhanced crystallinity of perovskite film, as shown in Figure 3a [42,43].

The influence of the generated 2D thin film on the optoelectronic properties of the perovskite films was explored using light absorption spectroscopy. The absorption profiles of the prepared films were investigated with the help of UV-vis absorption spectroscopy. We hypothesized that organic cations can minimize imperfection and trap states while also converting imperfections from 3D perovskite to 2D perovskite. Obtained spectra of the control and DABr-treated films, as shown in Figure 3c,d, reveal that the DABr-treated films exhibit comparatively lead in light absorption, with the DABr-2.5 being more robust in absorbing spectrum, followed by DABr-3.5 and DABr-1.5, respectively. The improved absorption characteristics of 2D-capping layer over 3D-MAPbI_3_ perovskite films will exaggerate the short circuit current density (J_SC_) of PSCs. As displayed in Figure 3d, the absorption edge of the DABr-treated film exhibits a slight blue shift compared with the control film, resulting from the generation of 2D perovskite and the introduction of Br into the film lattice structure [28]. It is well known that the bandgap of 2D perovskite is larger than that of 3D perovskite, thus the 2D/3D-stacked heterojunction will slightly enlarge the bandgap of perovskite film. In addition, the introduction of Br in perovskite film will make the valence band of the perovskite shift down, also leading to increase the perovskite bandgap. [20,21]. Furthermore, the light absorption in intensity of the treated film was improved compared to the non-treated film. This suggested the DABr post-treatment has a positive effect in light absorption ability of perovskite film. The high light absorption of the optimized DABr-treated perovskite films compared to the non–treated film can attribute not only to the increased crystallinity uniform perovskite with less defects.

In order to verify the hypothesis that the post-treatment of DABr perhaps leads to the variation in chemical bonding between I and Pb, X-ray photoelectron spectroscopy (XPS) was used to thoroughly investigate the effect of DABr on the chemical structure of control and DABr-2.5 perovskite films, as illustrated in Figure 3e,f and Appendix A. The peaks of I 3d and Pb 4f shifted to larger energy binding as the perovskite treated DABr-2.5, demonstrating that there is an interaction between DABr, Pb^2+^, and I^−^ [20,30,31,32]. Based on the above-mentioned analyses, we came forward with a credible proposed notion regarding the intermediary transitional process for the synthesis of high-quality MAPbI_3_-base film with larger grain size and less defect crystallinity [40,41,42,43].

To further ponder into the trap density of the prepared films, photo-physical measurements were measured with photoluminescence (PL) and time-resolved photoluminescence (TRPL), as shown in Figure 4. Both samples were deposited on glass side substrate of FTO, so as to explore the charge injection between the HTL and active layer [23]. The observed decays were analyzed by a bi-exponential model and the lifetime parameters were extracted. The TRPL decay curve was fitted by a bi-exponential model to calculate the decay time (*τ_i_*) of the control and DABr-2.5-treated films as Equation (1) and the average PL decay time (*τ_ave_*) was calculated by Equation (2) [20,23,27].
(1)$$f\left(t\right)=\sum_{i}A_{i}exp\left(-\frac{t}{\tau_{i}}\right)+f_{0}$$
(2)$$\;\tau_{ave}=\frac{\sum A_{i\;}\tau_{i}^{2}}{\sum A_{i\;}\tau_{i}}$$
where $A_{i}$ is the relative decay amplitude, $\tau_{i}$ is the decay time, and $f_{0}$ is the constant. The related parameters and the calculated carrier lifetime from TRPL are summarized in Appendix A. The decay curves were distributed into two decay parts, comprising a fast decay (*τ*_1_) and slow decay (*τ*_2_) [23,38]. The *τ*_1_ was attributed to the non-radiative interfacial recombination, and the *τ*_2_ was associated with the trap recombination in the perovskite bulk. The average carrier lifetime of the DABr-2.5-treated film was 145.36 ns, whereas the control layer exhibited a shorter average carrier lifetime of 63.34 ns. The extended carrier lifetime of the DABr-2.5-treated film (Figure 4a) indicates enhanced film luminescence due to the better carrier mobility, which is attributed to the reduced grain boundaries and trap sites as well [18,39,41]. Figure 4b exhibits the steady-state PL spectra of the pristine and DABr-2.5 films. In comparison with the control film, an elevated intensity peak with a justified blue shift in the PL spectrum of DABr-treated film is observed, supporting the fact that the treatment of DABr suppresses non-radiative recombination and widens the bandgap of perovskite film due to the incorporation of Br to the perovskite structure [20,26]. Such an improvement in the PL response of DABr-2.5 film could be a result of a well-engineered 2D-3D interface formed at the surface of 3D-MAPbI layers, leading to a decrease in statistics of the deep-level traps in bulk perovskite layers, which can greatly suppress the non-radiative recombination of charge carriers [36,40,44]. The corresponding film photoluminescence (PL) intensity mappings were performed to further investigate the effect of DABr treatment on the optical characteristic behaviors. The optical surface dynamics of the films are given in Figure 4c,d. Bright red color shows the PL intensity. PL intensity mapping provides a direct insight into the non-radiative exciton decay mechanism. The peak intensity of the control film is restricted by the bulk trap flux due to the poor crystal grain and poor carrier mobility [20]. However, the DABr-2.5 film shows a higher peak intensity, signifying a reduced non-radiative exciting decay.

Figure 5a illustrates the device architecture of the 2D-capped 3D MAPbI_3_ perovskite. 3D-MAPbI_3_ film treated with DABr-2.5 is employed as an active layer to prepare PSC with a 2D-3D hetero-structure film. As exhibited in Figure 5a, the device structure is FTO/C-SnO_2_/perovskite/spiro-OMeTAD/Au. A cross-SEM image of the final prepared device is shown in Figure 5b, demonstrating that the underlying ETL is consistently covered by the unceasing and vertically growth 3D MAPbI_3_ film with evenly distributed large-grains. To examine the photoelectric performance, the fabricated devices based on 3D-MAPbI_3_ and 2D-capped 3D-MAPbI_3_ films were measured under the forward and reverse scan directions, as presented in Figure 5c. The optimal PCE of the device based on 3D MAPbI_3_ reached 16.98% PCE with J_SC_ of 22.78 mA·cm^−2^, *V_OC_* of 1.02 V and FF of 72.75% under reverse scan, while achieving a 13.55% PCE under forward scan (Figure 5c). However, the device based on 2D-capped 3D film showed a comparatively enhanced performance with the highest PCE value of 19.10%, J_SC_ of 23.15 mA·cm^−2^, *V_OC_* of 1.06 V, and FF of 77.85% under reverse scan, while achieving a proximate PCE of 18.97% under a forward scan (Figure 5c). Obviously, DABr can inhibit the hysteresis and internal losses of PSCs. Additionally, to investigate the influence of different halides on the device photoelectric performance, the devices based on 3D-MAPbI_3_ with the post-treatment of different DA halides were fabricated. The J-V characteristic curves of the perovskite devices consisting of control, DABr-2.5, DAI-2.5- and DACl-2.5-treated films are shown in Appendix A and the results are presented in Appendix A. The device based on the 2D-3D film showed a significant improvement in the *V_OC_* and FF, indicating that the charge recombination is significantly restrained by the reduced defect density [36,43]. As expected and in the light of the above discussion, the treatment of DABr-2.5 not only outperformed the control device but also outperformed devices based on other DA halide-treated 2D-3D films.

Moreover, the statistical distribution of PCEs based on 30 individuals with control and DABr-2.5-treated perovskite films, as shown in Figure 5d, suggested the reliability and repeatability of the DABr-2.5 treatment method to produce 2D-capped 3D MAPbI_3_ PSCs. The statistical constriction of the DABr-2.5-treated device exhibited much finer spreading in the boosted efficiency regions in comparison to control devices. Furthermore, the incident photon to current efficiency (IPCE) curves and the integrated short circuit current density values based on the control and DABr-treated devices are presented in Figure 5e. Results showed a slightly but an obviously amplified spectral conversion profile with augmented integrated short circuit current density for the DABr-2.5 device in comparison to the control film-based device. Marginally improved spectral response and sound rise in integrated J_SC_ could be accredited to the morphological improvement of the stacked perovskite films caused by spectral sensitive 2D-capped layer homogenously encapsulating the vertically grown bottom 3D MAPbI_3_ layer [25,28,36]. Figurer 5f shows the steady-state output photo-current density and PCE curves of DABr-2.5-based devices at a bias of 0.90 V for 300 s. A steady-state output photo-current density of 21.05 mA cm^−2^ and the corresponding PCE of 18.25%. The above results indicated that DABr-based devices displayed a relatively high stable PCE curve and good reproducibility.

Moreover, the open-circuit voltage decay was measured in order to investigate the devices’ transient process of charge recombination mechanism. As shown in Figure 6a, the DABr-2.5-treated device exhibited a slower *V_OC_* decay behavior in comparison to the control device. The estimated decay constant of the DABr-2.5-treated device is 1.75 ns, which is larger than the control device (0.96 ns). As can be seen from Figure 6b, the electron lifetimes (*τ_n_*) can be calculated by using the following Equation (3) [18,21].
*τ_n_* = −*K_B_ Te*^−1^ (d*V_oc_*/d*t*)^−1^(3)
where *k_B_* represents the Boltzmann constant, *T* refers to the absolute temperature, and *e* is the elementary charge. Apparently, the DABr-2.5 device exhibits a longer *τ_n_* compared to the control device, indicating that the DABr-2.5 device has a lower charge recombination rate. *τ_n_* is the highest for each case at lower voltage region, particularly the device with DABr post-treatment, which is consistent with the *V_OC_* decay curve. The *τ_n_* of all the cases gradually decreases with the *V_OC_* increasing. However, the *τ_n_* of the DABr 2.5-based device remains higher than that of the control based, suggesting an improved film morphology [34,41].

Electrochemical impedance spectroscopy (EIS) studies were carried out to gain a better understanding of the charge dynamics and recombination mechanisms of both devices. As exhibited in Figure 6c, the charge recombination and electric properties of the devices were studied to explore the dynamics of charge carrier recombination at the interface between the perovskite and carrier transport layer. The equivalent circuit model illustrated in the inset of Figure 6c was applied to fit the data [45,46,47,48,49,50]. *R*_s_ stands for the series resistance and *R*_rec_ represents the recombination resistance. *R*_s_ is almost the same for the control and DABr-2.5-treated device, whereas the DABr-2.5-treated device shows a higher *R*_rec_ than that of the control device, indicating a lower recombination rate. This implies that DABr significantly suppresses recombination and enhances the carrier extraction for PSCs [20,51,52,53,54,55]. These EIS results demonstrated that the post-treatment of the perovskite film with DABr-2.5 enhaced the contact at interface between perovskite and HTL, and declined charge carrier recombination. The enhanced *R*_rec_ account is aligned with the observed incerased *V_OC_* and FF of the DABr-2.5 device [32,56]. Figure 6d shows the photo-graphs pictures of contact angles for control film and DABr-2.5-treated film. Obviously, the 2D film overlapping on the 3D film makes it more water-friendly and, by extension, more durable in relation to the amount of moisture in the air. The improvement in resistance against the humidity could probably help to improve the stability against the humidity [43,57]. The long-term stability has always been an urgent and difficult feature to achieve for PSCs. The stability of the aged devices based on 3D and 2D-3D films was tested under 30% RH at 25 °C. The PCE value of the device based on 3D film gradually reduced to nearly 20% after 700 h of aging, as exhibited in Figure 6e. In the meantime, the device based on 2D-3D film maintained 75% of the initial PCE value for 700 h. It is proved that the device based on 2D-3D film showed the resilient moisture stability, probably due to lack of any potential loophole in film morphology, which did not let any ambient species intercalate into the perovskite structure and decompose it. Consequently, it is concluded that the treatment of DABr for 3D perovskite film not only improve the device stability but can also refine the film crystal quality with large grains and fewer traps, which reduces the trap-assisted recombination and leads to a fine performance output.

## 3. Experimental Section

### 3.1. Materials and Reagents 

Fluorine-doped SnO_2_ coated glass (FTO glass) with 2.2 mm thickness and 15 Ω/sq resistance sheet were used for the PSCs, purchased from China. Lead iodide (99.99%), methylammonium iodide (CH_3_NH_3_I), diethylammonium bromide (DABr), BABr, phenyl-ethyl ammonium bromide (PEABr), and spiro-MeOTAD were bought from Xi’an polymer technology corp. *N*,*N*-dimethylformamide (DFM), dimethylsulfoxide (DMSO), chlorobenzene (CB), 4-Terstbutypridine (TBP), and lithium bis (trifluoromethyl sulphonyl) imide (Li-TFSI), were purchased from Aldrich (St. Louis, MO, USA). Acetylacetonate, anhydrous, titanium diisopropoxide and isopropanol (99.8%) were purchased from Acros Organics (New Jersey, NJ, USA). All the chemicals and solvents were stored in the glove-box before starting our experiment. All reagents were used as received. The SnO_2_ colloidal dispersion (tin(IV) oxide, 15% in H_2_O colloidal dispersion) was purchased from Alfa Aesar (Heysham, UK). Before use, the SnO_2_ solution was diluted by deionized water to 2.5%, followed by stirring at room temperature for 1 h.

### 3.2. Device Fabrication

FTO glasses were etched by using zinc power and hydrochloric acid (HCl). The obtained glass substrates were twice ultra-sonically cleaned with deionized water (DIW), liquid detergent solution, IPA, ACN and then ethanol. The FTO substrates were dried at the temperature 150 °C. The FTO were sintered at 500 °C for 35 min, in air to removed residual organic matter. All the substrates were UV Ozone cleaned for 15 min subsequently. The SnO_2_ precursor was spin-coated onto glass/FTO substrates at 5000 rpm for 30 s, and then baked on a hot plate in ambient air at 150 °C for 30 min. Then, films were cool down to room temperature. The perovskite MAPbI_3_ precursor solution was prepared by dissolving lead iodide (1.2 mol/mL) and methylammonium iodide (1.2 mol/mL) in an anhydrous solvent system of DMSO:DMF (1:4 of volume ratio). The deposited precursor solution was spin-coated initially at 1100 rpm for 10 s and then at 4500 rpm for 30 s during the second step. During the second stage of the spin-coating process, 100 µL of anhydrous chlorobenzene (CB) was dripped onto the center of the rotating film in 10 s prior to the end program. The obtained film was annealed at 60 °C for 2 min and 100 °C for 20 min. 50 μL of DABr in 2-isopropanol (IPA) solution with different concentrations (1.5 mg·mL^−1^, 2.5 mg·mL^−1^ and 3.5 mg·mL^−1^) was spin-coated on the surface of the prepared MAPbI_3_ film for 3 s and spin-coated at 4200 rpm for 15 s. The DABr post-treated films were thermally annealed at 100 °C for 15 min. For hole transport material, 73 mg of spiro-OMeTAD, 1 mL of CB, 17.5 μL of lithium bis (trifluoromethanesulfonyl) imide (Li-TFSI) solution (520 mg of Li-TSFI in 1 mL of acetonitrile) and 28 μL of 4-tert-butylpyridine were mixed. The 25 μL spiro-OMeTAD solution was spin-coated on the top surface of perovskite film at 4500 rpm for 30 s. Lastly, the gold (Au) electrode layer was deposited on the top of perovskite film, with 60 nm thickness by thermal evaporation technique under vacuum, at a constant evaporation rate of 0.6 nm/s.

### 3.3. Characterizations

A scanning electron microscope (Hitachi, Tokyo, Japan SU8010) was utilized to study the morphology. X-ray photo-electron spectroscopy (XPS) spectra were tested by a thermo fisher scientific ESCALAB 250XI (Shanghai, China). The crystallinity of perovskite film was measure by X-ray diffraction Cu Kα beam radiation of 0.15406 nm (X Pert Pro, Almelo, The Netherlands). Using UV-Vis spectrophotometer (SOLID 3700, SHIMADZU, Tokyo, Japan), the absorption spectra were measured. The steady state PL spectra of the produced samples were looked at using Edinburgh PLS980 (Edinburgh Instruments Ltd, Edinburgh, UK). Steady PL was recorded with a laser confocal Raman spectrometer (Princeton Instruments, Acton Standard Series SP-2558, Edinburgh Instruments Ltd, Edinburgh, UK) and a 532 nm laser using a home-built confocal microscope on a 10 × 10 μm^2^ sample area. TRPL measurement was recorded by the FLS980 steady-state/transient fluorescence spectrometer (Edinburgh Instruments Ltd, Edinburgh, UK). Samples were excited with a 488 nm pulsed diode laser with a repetition rate of 5 MHz and an excitation intensity of ~14 nJ/cm^2^, and the pulsed source was at 460 nm wavelength. The J-V characteristic curves were measured by utilizing forward (−0.1 to 1.2 V) or reverse (1.2 to −0.1 V) scans under AM 1.5 G solar illumination with a power intensity of 100 mW/cm^2^. The solar simulator equipped with (Keithley 2400) source meter (94043A, Oriel Instruments, Franklin, MA, USA) was calibrated with silicon cell in a nitrogen atmosphere. The cut-off voltage is set as (1.5 V) or (−1.5 V) for reverse or forward scans. 100 mA was used for the cut off current. The scan speed was 93 mV/s with a scan delay of 0 s. The solar cells were masked with a black cap of 0.09 cm^2^ area to evade the scattering light and to define the active region. The EIS data were fitted with help of ZSim-software version 3.20 with equivalent circuit. The EQE spectra was measured using 300 W Xenon Lamp with spectral resolution of 5 nm equipped with order sorting filter (Oriel Instruments, Franklin, MA, USA). A Zahner electrochemical workstation (Zahner Zennium, Kronach, Germany) was used for electrochemical impedance measurements (EIS) in the frequency range of 10 mHz to 1 MHz. Impedance data were analyzed using Z-view ZSim-software version 3.20 with equivalent circuit modeling software.

## 4. Conclusions

In summary, we developed a novel sequential deposition method for obtaining 2D-3D stacked perovskite film by the post-treatment of DABr. The introduction of DABr with an optimum proportion is benefit for obtaining a large grain and highly crystalline 2D-capped 3D-MAPbI_3_ perovskite film, effectively suppressing the nonradiative recombination at the interface between the perovskite and the charge transport layers. This method took the perovskite post-treatment strategy to the next level and offered a simple way for the fabrication of hetero-structure perovskite film with some outstanding photovoltaic performance and resilient stability.

## Figures and Tables

**Figure 1 molecules-28-01592-f001:**
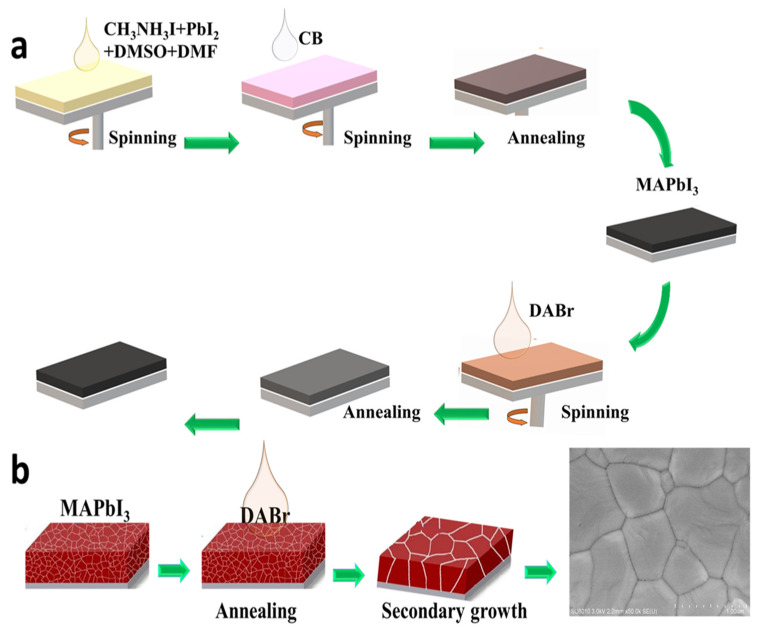
(**a**) Illustration for the fabrication of DABr capped 3D MAPbI_3_ film via DABr surface treatment method, (**b**) pictorial presentation of DABr capped MAPbI_3_ film evolution via DABr surface treatment method.

**Figure 2 molecules-28-01592-f002:**
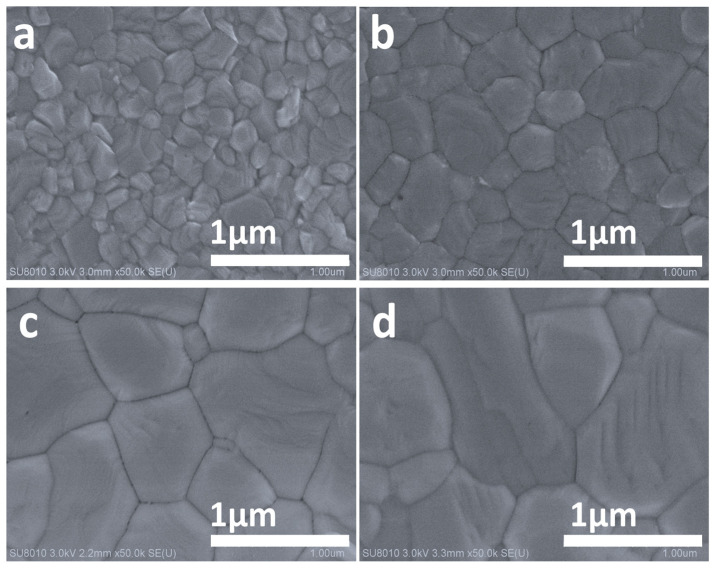
Top view of SEM images for (**a**) control, (**b**) DABr-1.5, (**c**) DABr-2.5 and (**d**) DABr-3.5-treated 2D-3D perovskite films.

**Figure 3 molecules-28-01592-f003:**
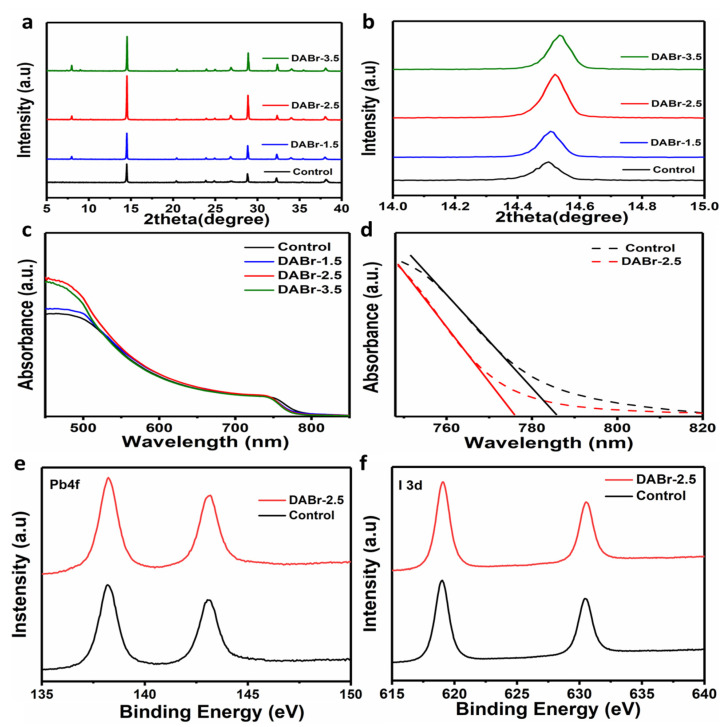
(**a**) X-ray diffraction (XRD) patterns and (**b**) extended XRD region of the films proccessed with different concentrations of DABr solution. (**c**) UV-vis absorption spectra of the control film and the films treated with different concentrations of DABr, (**d**) fingerprint region of UV-vis absorption spectra of the films. X-ray photoelectron spectroscopy (XPS) spectra of (**e**) Pb 4f and (**f**) I 3d for control and DABr-2.5-treated films.

**Figure 4 molecules-28-01592-f004:**
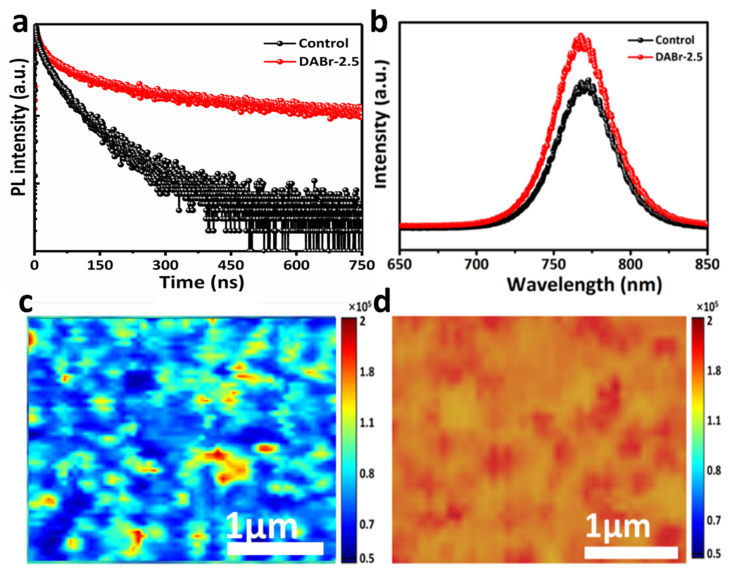
(**a**) Time-resolved photoluminescence (TRPL) and (**b**) steady-state photoluminescence (PL) of the control and DABr-2.5 films deposited on glass. (**c**) Single-point PL spectral intensity mappings images of the control and (**d**) DABr-2.5 films.

**Figure 5 molecules-28-01592-f005:**
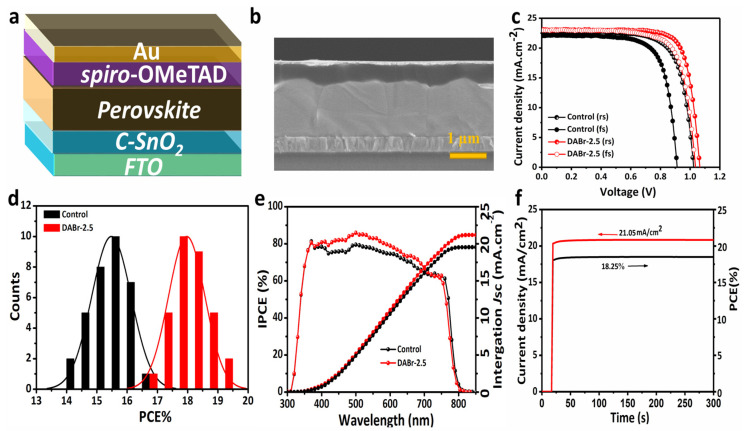
(**a**) Schematic diagram of the planar prepared device. (**b**) Cross−sectional SEM image of the final device based on DABr−2.5. (**c**) Current density-voltage (J−V) curves scanned under forward and reverse bias for prepared devices based on the control and DABr−2.5−treated films. (**d**) The statistical distribution of PCEs recorded for 30 tested devices for each type. (**e**) Incident to photo-conversion efficiency (IPCE) spectra of devices with the control and DABr−2.5−treated films. (**f**) Stable current output and PCE curves of the DABr−2.5−based device at the maximum power point.

**Figure 6 molecules-28-01592-f006:**
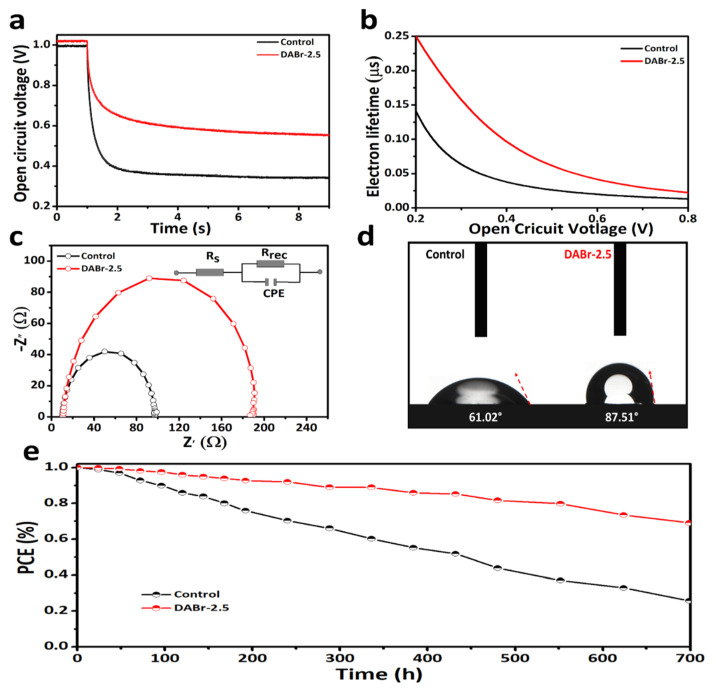
(**a**) Open−circuit voltage decay curves, (**b**) electron lifetimes curves (**c**) electrochemical impedance spectroscopy (EIS) curves of devices based on the control and DBAr−2.5−treated films. (**d**) Contact angle photographs of the control and DABr−2.5−treated films. (**e**) The normalized PCEs of the prepared devices aged for 700 h under 30% RH at 26 °C.

## Data Availability

Data is contained within the article or Appendix A.

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
