# Peer review of "Graded 2D/3D Perovskite Hetero-Structured Films with Suppressed Interfacial Recombination for Efficient and Stable Solar Cells via DABr Treatment"

_molecules, 2023, doi:10.3390/molecules28041592_

Round 1
Reviewer 1 Report
In this study, authors introduced diethylammonium bromide (DABr) as a post-treatment material on the MAPbI3 film to obtain high-quality two-dimensional/three-dimensional (2D/3D) stacked hetero-structure perovskite solar cells. The surface defect density of MAPbI3 film is reduced by forming a 2D DA2MAn-1PbnX3n+1 capping layer on the surface of the MAPbI3 layer. This further suppressed the charge recombination at the interface between the perovskite and charge transport layers. As a result, the PCE of 19.10 % with a significant improvement in the VOC parameter was achieved with a prolonged stability. The manuscript is well-written and the analysis is solid. However, it is already well-known that 2D perovskites can improve solar cell stability. It looks like that the authors just used a routine approach to improve the device performance. To highlight the novelty of this study is required. Authors should address the following queries:
1- Recently, DABr-doped MAPbI3 is reported in the literature (10.1021/acsami.0c01672), please explain the main differences of your work with this one or other similars in the literature.
2- The Voc in comparison to works in the literature is lower. If the 2D layer improves the interfacial transport, I would imagine the Voc to be higher.
3- Normally, 2D perovskite capping layers block the charge transfer because the insulating behavior of the large organic cations. Why did the solar cells with DA2MAn-1PbnX3n+1 layer even have higher PL quenching efficiency and higher Jsc?
4- The authors show the comparison on long-term moisture stability of the devices. How about the heat stability?
5- Figure 5b is not clear because of the coloring of the layers.
6- Authors stated the functional approaches to improve the efficiency as “…according to the optimized material stoichiometry, solvent engineering, sequential deposition technique,12−17 interfacial engineering and additive engineering.18−25…”. However, some references are missing in this statement. Some suggestions from recent studies: (10.1016/j.nanoen.2021.106157 and 10.1002/adfm.202102124 for solvent engineering, 10.1021/acsami.0c17893 for interfacial engineering)
7- Regarding the 2D/3D studies, some recents studies should be cited and discussed in the main text
8- SEM images of target films in lower magnification should be provided to clearly see the grain distrubiton and surface morphology.
9- How thick can the 2D perovskite layers be, before limiting efficiency?
Reviewer 2 Report
The authors presented "Graded 2D/3D Perovskite Hetero-Structured Films with Suppressed Interfacial Recombination for Efficient and Stable Solar Cells via DABr Treatment" by exploring all possible methods. The idea is novel and it has significant importance in the research community. However, before publishing in the journal, I recommend the following major revision which may improve the impact of the study.
1- The efficiency of the treated sample improved because of suppressed interfacial recombination. However, apart from TRPL, there is no such direct spectroscopic evidence mentioned. Transient absorption is one of the main tools to identify those processes. If possible the author may include this study.
2- In Fig 3a the authors observed a peak below 10-degree for the treated samples and mentioned that it is due to the 2D phase. To support this they cited ref 35 and 39. However, in ref 39 the XRD is presented from 10-degree. Please comment on it.
3- It is mentioned that the absorption is improved of the treated (2.5) sample with blue shifting in energy due to 2D phase. It is most likely because of concentration otherwise the effect will be more for 3.5 samples. Authors must comment on it. Moreover, the possible of ion migration is not discussed, which plays a vital role in halide perovskite.
4- The authors must mention the excitation and monitoring wavelength for the TRPL measurements. Also, it is important to show the fitting from where the lifetime was calculated. The offset of the rising of the traces must fix to zero.
5- Do the authors confirm which colors in the 2D plots in Fig 4 c and d represent the PL of the materials? What is excitation fluence?
6- How the electron lifetime in figure 6b is calculated is not clear. It needs a better explanation.
7- This article missed to cite all other relevant literature. Authors are encouraging to follow similar other literature DOI: 10.1021/acs.jpcc.6b07876; DOI: 10.1039/d0tc05010e etc.
Round 2
Reviewer 1 Report
Authors addressed the queries, I suggest the publication of this study in Molecules.
Reviewer 2 Report
The authors addressed my concerns. Now, this form may accept for publication.